# The AGE-RAGE Axis and the Pathophysiology of Multimorbidity in COPD

**DOI:** 10.3390/jcm12103366

**Published:** 2023-05-09

**Authors:** Niki L. Reynaert, Lowie E. G. W. Vanfleteren, Timothy N. Perkins

**Affiliations:** 1Department of Respiratory Medicine, School of Nutrition and Translational Research in Metabolism, Maastricht University Medical Center+, 6229 ER Maastricht, The Netherlands; 2COPD Center, Department of Respiratory Medicine and Allergology, Sahlgrenska University Hospital, 413 45 Gothenburg, Sweden; 3Department of Internal Medicine and Clinical Nutrition, Institute of Medicine, Sahlgrenska Academy, University of Gothenburg, 405 30 Gothenburg, Sweden; 4Department of Pathology, School of Medicine, University of Pittsburgh, Pittsburgh, PA 15261, USA

**Keywords:** COPD, multimorbidity, advanced glycation end products, receptor for advanced glycation end products, chronic inflammation, aging

## Abstract

Chronic obstructive pulmonary disease (COPD) is a disease of the airways and lungs due to an enhanced inflammatory response, commonly caused by cigarette smoking. Patients with COPD are often multimorbid, as they commonly suffer from multiple chronic (inflammatory) conditions. This intensifies the burden of individual diseases, negatively affects quality of life, and complicates disease management. COPD and comorbidities share genetic and lifestyle-related risk factors and pathobiological mechanisms, including chronic inflammation and oxidative stress. The receptor for advanced glycation end products (RAGE) is an important driver of chronic inflammation. Advanced glycation end products (AGEs) are RAGE ligands that accumulate due to aging, inflammation, oxidative stress, and carbohydrate metabolism. AGEs cause further inflammation and oxidative stress through RAGE, but also through RAGE-independent mechanisms. This review describes the complexity of RAGE signaling and the causes of AGE accumulation, followed by a comprehensive overview of alterations reported on AGEs and RAGE in COPD and in important co-morbidities. Furthermore, it describes the mechanisms by which AGEs and RAGE contribute to the pathophysiology of individual disease conditions and how they execute crosstalk between organ systems. A section on therapeutic strategies that target AGEs and RAGE and could alleviate patients from multimorbid conditions using single therapeutics concludes this review.

## 1. Introduction

Activation of the immune system is critical during phases of acute inflammation (in response to infection or injury) to neutralize foreign agents and restore homeostasis. However, when the activation of the immune system goes unchecked, a constant and unnecessary inflammatory response follows and may result in damage to the affected tissues and organs [1,2]. Such inflammation is the major pathophysiological mechanism that leads to organ dysfunction in a number of non-communicable, chronic inflammatory conditions, including autoimmune disease, cancer, cardiovascular disease (CVD), and lung diseases. Collectively, these conditions are considered the most significant cause of mortality worldwide and often occur simultaneously [3]. A key example is chronic obstructive pulmonary disease (COPD), a lung disease driven by exaggerated inflammation, most commonly caused by cigarette smoking. COPD is considered to be the third leading cause of death worldwide [4]. Patients with COPD are often multimorbid, as they commonly suffer from multiple chronic (inflammatory) conditions or comorbidities [5]. In the modern world, chronic inflammatory conditions are caused by genetic predisposition, environmental irritants, a modern lifestyle, and most importantly, aging [1]. Advanced glycation end-products (AGEs) are a heterogeneous group of molecules formed by various chemical reactions and molecular rearrangements. AGEs can accumulate in the body through exogenous sources, such as food consumption and cigarette smoke, or by endogenous formation due to high levels of circulating glucose in diabetes or through aging [6]. AGE accumulation can have detrimental effects in various tissues, as AGEs can cause inflammation and alter the functions of proteins. AGEs are well-known to cause adverse inflammatory reactions in COPD and other chronic inflammatory conditions [7]. The receptor for advanced glycation end-products (RAGE) is a member of the immunoglobulin superfamily of receptors that was discovered and named as the first known receptor for AGEs [8]. RAGE binds to a plethora of ligands and plays a central role in promoting inflammation in various chronic conditions [7,9]. Ligand-induced activation of RAGE promotes transcription of pro-inflammatory genes, RAGE ligands, as well as RAGE itself [10,11]. This feed-forward positive feedback mechanism perpetuates persistent and damaging inflammation, which contributes to the manifestation of chronic inflammatory diseases such as COPD and comorbidities (Figure 1) [7,12]. In this review, we discuss the AGE-RAGE signaling axis as a driver of inflammation, its contribution to the pathophysiology of multimorbidity in COPD, and its potential therapeutic implications.

## 2. COPD

COPD is a lung disease primarily defined by airflow obstruction that is not fully reversible. COPD often worsens over time due to the progressive development of abnormalities of the conducting airways (bronchitis or bronchiolitis) and/or alveolar space (emphysema) [13]. Patients present with symptoms of dyspnea and chronic cough, with or without sputum production. In particular, chronic, persistent airway inflammation promotes airway remodeling and destruction of lung parenchyma, leading to loss of lung function [14]. At this point, structural damage and loss of lung function cannot be reversed, although progress is being made in our understanding of lung cell progenitor functions in settings of injury and repair and how their dysfunction contributes to tissue remodeling [15]. Currently, most treatment options are aimed at reducing symptoms and preventing acute disease exacerbations. No interventions have been shown to prevent disease progression, and smoking cessation can only slow progression.

Cigarette smoking is the main injurious trigger and risk factor for the development of COPD in Western countries [15,16,17]. In addition, COPD also occurs in non-smokers and is associated with occupational exposures to dust and gases, as well as environmental exposure to air pollution. In developing countries, exposure to household air pollution from biomass smoke (the incomplete combustion of organic materials used for cooking and heating) is thought to be responsible for COPD development [18]. In non-cigarette smoke COPD, the phenotype involves increased blockage of small airways. It is characterized by eosinophilic inflammation that is often more responsive to bronchodilators. On the other hand, this phenotype is more likely to be steroid-refractory but involves a slower rate of decline in lung function. It occurs more frequently in females with a higher BMI and at an earlier age [19]. GWAS studies have identified various genetic susceptibility loci. This, in conjunction with a range of epigenetic markers, explains why not all smokers develop COPD. In addition, alpha-1 antitrypsin deficiency is the only monogenetic cause of emphysema, which leads to early disease onset (<55 years of age) in the absence of smoking [15].

COPD is typically diagnosed after the age of 65. COPD is thereby considered an age-related condition, representing an accelerated decline in lung function. In recent years, however, early life events that negatively alter the trajectory of normal lung function development have been increasingly recognized as a cause of COPD [20].

Regardless, COPD shares disease pathways with normal aging, and most of the aging hallmarks that were described by Lopez-Otin in 2013 are present in and contribute to COPD [21,22]. One of these hallmarks is altered intercellular communication in the form of low-grade systemic inflammation [14]. Although the origin of systemic inflammation is debated, it relates to the development of symptoms beyond the lungs. These extra-pulmonary manifestations include fatigue, sarcopenia, anxiety, and depression, as well as cardiovascular disease (CVD), which contribute to overall disease burden and mortality.

## 3. Multimorbidity

COPD is an example of the epidemic of non-communicable chronic inflammatory diseases that put an enormous burden on societies worldwide. The increased lifespan of the human species and an increasingly unhealthy lifestyle are the main causes of the growing prevalence of chronic inflammatory disorders. Importantly, the elderly often suffer from multiple chronic inflammatory diseases, termed multimorbidity, in part because of shared risk factors. Multimorbidity intensifies the burden of individual diseases, further impacts quality of life, and complicates disease management.

Multimorbidity is very frequent in patients with COPD; conversely, COPD is often a comorbidity with other chronic diseases. Almost all COPD patients suffer from at least one other chronic disease, the majority of whom suffer from several. Examples of comorbidities that are most prevalent in COPD include hypertension (17–64.7%), coronary artery disease (19.9–47.8%), diabetes (10.2–45%), osteoporosis (6.9–20.1%), depression/psychiatric disorders (12.1–33%), obesity (2.8–20%), and chronic kidney disease (9.9–25.8%) [5]. Skeletal muscle dysfunction and low muscle mass are also highly prevalent in COPD [23]. Often, these comorbidities go unrecognized and untreated, despite their importance [24]. COPD patients suffering from comorbidities typically have worse outcomes, such as impaired quality of life, increased exacerbations, more frequent hospital admissions (including those with a longer duration), and a worse response to therapy. Moreover, heart failure, diabetes, hyperlipidemia, hyperuricemia, osteoporosis, and sleep apnea were found to strongly contribute to COPD progression [25]. Furthermore, a recent meta-analysis found increased mortality in COPD to be associated with coronary artery disease, diabetes, psychiatric disorders, chronic heart failure, atrial fibrillation, and other arrhythmias [5]. Importantly, the main cause of death in mild-to-moderate COPD patients is CVD. On the other hand, reduced forced expiratory volume in one second (FEV_1_) is a marker of premature death from all causes and not just a marker of lung function [26]. These epidemiological data illustrate the clinical importance of the complex interplay between disorders affecting different organs.

Epidemiological studies have shown that certain diseases are more likely to occur together in the same individual. Therefore, comorbidities appear in clusters. A systematic review identified 97 clusters of comorbidities, of which the cardiometabolic, mental health, and musculoskeletal clusters were the most prominent [27]. Interestingly, COPD was present in all three clusters. Unbiased analysis of various chronic inflammatory disorders using COPD as an index disease revealed five clusters of comorbidities: a less comorbid cluster, a cardiovascular cluster, a metabolic cluster, a psychological cluster, and a cachectic cluster [28]. In the HUNT study, similar clusters were identified: one with fewer comorbidities, a psychological cluster, a metabolic cluster, a cachectic cluster, and a cluster with myocardial infarction and renal failure. The psychological and cachectic clusters displayed an increased mortality risk, a lower quality of life, and more exacerbations over a 16-year follow-up [29]. The psychological and cachectic clusters were also found to be COPD-specific [23].

The fact that comorbidities cluster together highlights the possibility that they are driven by shared underlying molecular networks. Network analysis is used to identify common genes, proteins, and signaling pathways shared between COPD and comorbidities. This approach has shown that commonalities and coexistence stem in part from genetic susceptibility and exposure to stressors, most notably tobacco smoke [30,31]. Differential connections with and within comorbidity networks also depend on clinical variables and characteristics, such as the presence of chronic bronchitis, the severity of airflow limitation, age, or BMI [31,32].

Shared risk factors of COPD and comorbidities trigger common pathways and explain to a large extent the co-occurence of these diseases. These factors include age, smoking, physical inactivity, a poor diet, obesity, and low socio-economic status. Most of these risk factors contribute to chronic, low-grade systemic inflammation, which is known to accelerate aging in various organs through inflammaging (Figure 2).

In this context, COPD and its multimorbid spectrum can be understood as the result of the accumulation of gene-environment interactions encountered by an individual over the course of their life. Importantly, because the biological responses to and clinical consequences of different exposures might vary according to both the age of an individual at which a given gene-environment interaction occurs and the cumulative history of previous gene-environment interactions, a time axis should be integrated in pathogenic models of COPD and comorbidity [33].

## 4. RAGE

### 4.1. RAGE Signaling Leads to Chronic Inflammation

RAGE has a central role in amplifying inflammatory responses, and an increasing number of studies link RAGE to diseases characterized by underlying chronic inflammation. RAGE expression is generally low in adult organs under basal conditions, with its highest expression in the lung [34]. It is upregulated in inflammatory settings and with increasing age. Importantly, RAGE ligands also typically accumulate in tissues during inflammation, aging, or other types of tissue stress.

It is now known that, in addition to AGEs, a multitude of damage-associated molecular patterns (DAMPs) bind and activate RAGE. These include S100 calgranulins and the DNA-binding protein high mobility group box-1 (HMGB1)/amphoterin [8,35,36,37], amyloid *β*-peptides/fibrils [38], and complement C3A [39]. Recently, RAGE has also been shown to interact with various forms of nucleic acids. For instance, RAGE mediates the uptake of CpG nucleotides, which are delivered to TLR9, promoting downstream inflammatory responses [40,41]. Intracellularly, RAGE interacts with various adaptor proteins, the best studied of which is diaphanous-related formin 1 (DIAPH1). Upon ligand binding, signaling through these adapter proteins initiates a range of signaling cascades, including MAPK, which induce activation of various transcription factors, including NF-κB, STAT3, SP-1, CREB, AP-1, and EGR-1. In turn, these stimulate biological processes that affect inflammation, cell survival or death, migration, proliferation, and differentiation. RAGE activation also promotes the production of reactive oxygen species (ROS) through NADPH oxidases (Figure 3).

After ligand binding, the receptor is internalized and recycled. Importantly, RAGE activation also leads to the upregulation of RAGE expression itself. NF-κB plays a pivotal role herein [42]. Moreover, RAGE ligands activate NF-κB in a prolonged and sustained manner by circumventing negative feedback loops [43]. As RAGE signaling also upregulates many of its ligands, a vicious cycle of self-propagating inflammation is created. Complex formation between ligands, such as between HMGB1 and DNA, can increase RAGE stimulation [40]. Lastly, RAGE cooperates with TLR4 and TLR2 since they share common ligands [44].

In addition to the classical signal transduction routes that arise from membrane-bound RAGE, a number of studies have reported nuclear accumulation of RAGE [45,46]. Importantly, the functionality of nuclear RAGE was recently demonstrated. It was shown that RAGE is recruited to sites of DNA damage (double strand breaks) and that it is phosphorylated by ATM kinase, leading to the activation of ATR kinase signaling and the prevention of senescence (Figure 3) [47]. Interestingly, other membrane-located pattern recognition receptors, such as TLR4, have also been shown to function in DNA damage repair [48].

### 4.2. RAGE Structure and Variants

RAGE is composed of three extracellular ligand-binding domains: two constant domains (C1, C2) and a variable (V) immunoglobulin domain that includes two glycosylation sites. These three domains provide the molecular basis for binding multiple ligands. A transmembrane domain elicits ligand-induced oligomerization, and a cytoplasmic domain interacts with the various downstream effectors detailed above (Figure 3).

In addition to full-length RAGE, many different isoforms can be expressed (Figure 4), several of which are truncated forms. Alternative splicing, leading to deletion of the transmembrane and cytosolic domains, gives rise to so-called endogenous soluble RAGE (esRAGE or RAGE V1) [49], which acts as a ligand scavenger. Overexpression of esRAGE can indeed blunt some effects elicited by RAGE [50]. A dominant negative (DN) variant lacks only the cytoplasmic domain, thereby lacking intracellular signaling and blunting ligand responses. An N-truncated variant, on the other hand, lacks the AGE-binding V-domain. Human RAGE splicing is highly tissue-dependent; full-length RAGE expression is very abundant in the lungs and aortic smooth muscle cells, while esRAGE mRNA is most prevalent in endothelial cells [51]. Moreover, different ligands upregulate different RAGE isoforms [52]. The extracellular domain of RAGE can also be released as a soluble variant by proteolytic cleavage by A Disintegrin, a metalloproteinase domain-containing protein (ADAM)10, and matrix metalloproteinases (MMP)3, 9, and 13 [53,54,55,56]. This cleaved form is termed sRAGE. Experiments in which high doses of exogenous sRAGE were administered have proven its scavenging properties [57,58]. Endogenous levels of sRAGE are, however, much lower, making it unclear whether it exerts this function in physiological settings. It is also unclear whether the scavenging properties of esRAGE are distinct from those of sRAGE. Of note, in circulation, only 20 percent of the total sRAGE levels are attributed to esRAGE [59]. Lastly, clearance of sRAGE from the plasma by the kidneys is an important determinant of the circulating levels.

Next to isoforms, a variety of polymorphisms have been described that modulate RAGE expression and/or functionality (Figure 4). For instance, both the -374T/A (rs1800624) and -429T/C (rs1800625) SNPs located in the *AGER* promotor lead to upregulated expression of *AGER* [60,61]. A substitution of glycine by serine at amino acid position 82 (G82S, rs2070600) enhances ligand binding and inflammatory reactions [62]. The consistent glycosylation at asparagine 81 resulting from the substitution is responsible for the improved and sustained ligand binding [63]. In a Korean population-based study, this SNP was also positively associated with elevated levels of serum AGEs, C-reactive protein (CRP), and TNFα [64]. Importantly, this polymorphism is also associated with lower sRAGE levels in vitro and in vivo without impacting *AGER* gene expression [64,65]. Lower sRAGE levels have also been connected to a polymorphism at the splice site that leads to esRAGE (-6G/A, rs2071288) [66,67].

### 4.3. Pulmonary RAGE

In the physiological state, pulmonary tissue displays high expression levels of RAGE compared to other organs [68]. High baseline expression is especially prominent in the basolateral membrane of type 1 alveolar epithelial (ATI) cells. RAGE mediates cellular adherence to collagen IV, interactions between adjacent cells, cell spreading, and the transition of alveolar type 2 (ATII) cells to ATI cells, which facilitate gas exchange [69,70,71]. In line with the appearance of ATI cells, the expression of RAGE in the lungs increases during gestation [65]. Furthermore, minor expression of RAGE in the lung is present in ATII cells, bronchial epithelial and smooth muscle cells, vascular endothelial cells, and alveolar macrophages [72,73]. The findings that the lungs of *AGER* knockout mice are more compliant and display deficient alveolarization, fewer radial alveolar counts, an increased mean linear intercept, thicker alveolar walls, and a leaky alveolar-capillary membrane support the essential role of RAGE in alveolarization and in mature alveoli [74,75,76]. *AGER* knockout mice also spontaneously develop features of lung fibrosis with late age [77] and display an increase in DNA damage and senescence, which leads to an increased formation of lung carcinomas over time compared to wild-type animals [47]. These data are in line with the proposed function of RAGE in DNA repair attributed to its nuclear localization. Surprisingly, however, *AGER* knockout mice are protected from bleomycin-induced lung fibrosis [78]. Embryonic overexpression of *AGER* in ATII cells was found to cause severe lung hypoplasia and perinatal morbidity [79,80]. Hemizygous *AGER* transgenic mice presented with impaired alveolar morphogenesis, which was associated with fragmented elastin, increased cell death, and diminished proliferation [81]. These animal studies indicate that germline or embryonic disruptions of physiological RAGE expression levels lead to impaired pulmonary development and adverse outcomes.

In various lung diseases, RAGE is upregulated and associated with inflammation and oxidative stress [82]. A number of its ligands are also elevated in lung disorders. The AGE CML in particular was found in cells in which RAGE was also upregulated, namely in bronchial epithelial cells, ATII cells, alveolar macrophages, and some endothelial cells [83].

### 4.4. Genetics as a Common Cause of Altered RAGE Signaling

Elevated RAGE expression is common in chronic inflammatory diseases and is causally involved in the maintenance of the chronic state of inflammation. In addition to the ligand-mediated induction of RAGE expression, genetic variations in the *AGER* gene are associated with increased susceptibility to the development of these diseases. For some polymorphisms, this could be attributed to enhanced signaling arising from the variant forms of RAGE. Importantly, some genetic polymorphisms are implicated in various diseases and share common associations with circulating sRAGE levels. For an excellent overview, see [84].

The G82S SNP is the most researched genetic variant in *AGER*. It is associated with lower plasma sRAGE levels and higher circulating levels of inflammatory mediators and markers of oxidative stress [64]. Two independent GWAS studies in healthy individuals of European ancestry reported a significant association between G82S and lung function measures deviating from normal [85,86]. This SNP is also associated with the presence of COPD and emphysema [67,87,88]. Moreover, it is associated with increased insulin resistance and complications of type 2 diabetes, including retinopathy, nephropathy, and microangiopathy, as well as type 1 diabetes. However, not all studies in diabetes could reproduce these findings, with ethnicity being a likely confounder. Similarly, conflicting findings are reported on the association of G82S with CVD depending on the ethnicity of the studied population [84]. Other SNPs are studied less often, and data are lacking to commonly link them to multiple diseases.

### 4.5. Early Origins of Altered RAGE Signaling across Diseases

Previously, research into the development of chronic, age-related diseases focused on the decay of organ function after adulthood. In recent years, however, it has become apparent that these diseases can have roots in early life. Early life events that negatively affect organ growth and maturation prior to and after birth can result in a lower peak functioning of organs that is reached in late adolescence. From this point on, the threshold of organ malfunction and symptom development will be reached more rapidly. In COPD, for example, it is postulated that a substantial portion of patients develop the disease because of early life events that hinder normal lung development and lead to submaximal lung function attainment in early adulthood [20]. Importantly, commonalities exist between early-life susceptibility factors and factors that drive accelerated lung function decline in later life. These factors include, for example, a poor maternal diet, stress, and smoking. Fetal exposures not only affect in utero organ development in a broad sense but also constitute an increased risk for premature birth. Together, these very early exposures set an individual on a suboptimal course of life.

For the lungs, an extremely low birth weight (<1000 g) and being born extremely preterm (<28 weeks) constitute a risk for reduced respiratory function and exercise capacity in later life [89]. Attainment of low peak lung function in early adulthood is not only associated with a higher incidence of respiratory conditions but also with cardiovascular and metabolic diseases in later life [90]. In addition, prematurity itself also increases the risk for chronic kidney disease [91], heart failure [92], increased low-density lipoprotein (LDL) levels and high blood pressure [93,94], stroke, diabetes, and asthma [95].

Given the essential role of RAGE in normal lung development, it is not surprising that studies have shown RAGE aberrations in children that are born prematurely. For instance, bronchoalveolar lavage fluid (BALF) sRAGE levels are reduced in infants born prematurely and at an increased risk of developing bronchopulmonary dysplasia (BPD) [96]. In addition, attenuated cord blood sRAGE levels can serve as predictors of BPD [97]. However, it remains to be determined whether low sRAGE levels reflect the underdevelopment of the lungs (i.e., lower numbers of ATI cells) or result from attenuated RAGE expression or shedding. Additionally, lower sRAGE levels could contribute to the inflammatory state of the preterm lungs as well as other organs negatively affected by preterm delivery. Regardless, it should be taken into consideration that interventions in the neonatal intensive care unit, including oxygen supplementation and ventilation, can affect RAGE signaling and sRAGE levels as well.

Smoking is known to induce epigenetic aberrations that disturb normal lung development and accelerate lung function decline. Therefore, a recent study examined whether maternal smoke exposure and COPD share common divergent methylation patterns and consequent downstream gene expression. The AGE-RAGE axis was identified as a pathway with common differential methylation and consequent altered gene expression in an in utero smoke exposure model and in COPD [98]. This study indicates that the AGE-RAGE axis is commonly affected in COPD and in an early-life exposure model that predisposes to COPD. This suggests that maternal smoke-induced methylation of targets in the AGE-RAGE pathway during pregnancy could predispose to the development of COPD. In line with this, it was reported that smoking during pregnancy increases RAGE expression [99], which persists into adulthood in animal models [100]. Moreover, combined prenatal and perinatal nicotine exposure increased cord length, which was associated with enhanced RAGE and NF-κB expression [101]. It remains to be determined whether maternal smoking affects RAGE in other organs in the short and long term and whether it synergizes with the effects of smoking in adulthood to increase the risk of developing other diseases.

## 5. Common Causes Leading to AGE Accumulation

### 5.1. Conditions Leading to Enhanced Endogenous AGE Formation in Chronic Inflammatory Disease and Aging

AGEs are a heterogeneous group of stable compounds formed non-enzymatically from various precursors through chemical reactions and molecular rearrangements (Figure 5). In brief, the carbonyl group of a reducing sugar condenses with a free amine group, which, after rearrangements and further reactions, leads to irreversible modification of the targeted macromolecule. For a detailed overview of the processes and reactions leading to AGE formation, we refer the reader to additional articles [6,102]. We focus in this review on protein modifications by AGEs, as they are the most studied targets with respect to their formation and consequent effects.

AGEs are heterogeneous in many aspects, including differences in the various precursors they are derived from and the reactions through which they are formed. They can also be categorized based on chemical and physiological properties [103]. Some AGEs, for instance, have fluorescent properties (e.g., pentosidine); some are able to induce protein crosslinking (e.g., pentosidine; glyoxal-lysine dimer, or GOLD; methylglyoxal-lysine dimer, or MOLD); whereas others are non-fluorescent and not capable of inducing crosslinking (e.g., *N*^ε^-(Carboxymethyl)lysine (CML) and *N*^ε^-(Carboxyethyl)lysine (CEL)). They also vary in molecular weight, from single amino acids to peptides to proteins.

Endogenous formation of AGEs is a very slow process, arising as byproducts of cellular carbohydrate metabolism. This stable, irreversible posttranslational modification occurs more rapidly under conditions of oxidative stress and in the presence of high glucose and fructose levels through the formation of reactive (di)carbonyl intermediates. Hyperlipidemia also accelerates AGE formation through lipid peroxidation products such as malondialdehyde (MDA) and 4-hydroxynonenal (HNE) (Figure 5). Because hyperglycemia is an important driver of their formation, AGEs have been studied most thoroughly in diabetes and its complications. However, they likely play a pivotal role in most, if not all, chronic inflammatory disorders since chronic inflammation and associated oxidative stress also provide conditions conducive to their enhanced formation. In addition, AGEs stochastically accumulate with advancing age, and a key enzyme in AGE formation, aldose reductase, is increased in aging tissues, which can be involved in the increased formation of AGEs during aging [104]. AGEs have been shown to play a role in both normal aging and the accelerated aging that is observed in many chronic inflammatory diseases. When AGEs target long-lived proteins, such as those present in the extracellular matrix (ECM), they can serve as a biomarker for their life-long accumulation and are considered markers of biological aging. The levels of AGEs in the skin can be assessed non-invasively by measuring skin autofluorescence, which correlates with chronological and biological age [105].

### 5.2. Enhanced Intake through a Western Diet

Diet plays an important role in the induction and progression of chronic low-grade inflammation in many chronic diseases. A Western diet rich in processed and red meats, high-fat dairy, refined grains, sweets, and desserts contains high levels of AGEs and induces endogenous AGE formation. This contributes to the accumulation of AGEs in our bodies, which is therefore an important potential contributor to Western lifestyle-associated diseases.

The chemical reactions that lead to the formation of AGEs were actually first described in the browning of proteins during baking or broiling by Maillard in 1912. Their formation is especially high when protein-rich and sugar-rich food is thermally processed, which provides food with taste, aroma, and texture. As an initial attempt to quantify exposures to AGEs through the diet, the CML content of a large list of foodstuffs and various cooking methods used have been documented [106,107,108]. Although this does not capture the full spectrum of dietary AGEs, it represents an important starting point for research into their adverse effects and can be used to develop dietary guidelines. AGEs are highly prevalent in bread, coffee, processed meat, and even infant formula. Conversely, fruit and vegetables contain very low levels of AGEs. Moreover, although most beverages are low in AGEs, sugared drinks, especially those sweetened with sucrose and high-fructose corn syrup, drive the endogenous production of AGEs [109].

Initially, it was believed that AGEs were poorly absorbed, but it is now established that about 10% are absorbed through the intestine [108]. Despite the limited absorption of dietary AGEs, various animal studies using diets enriched in AGEs have shown that they accumulate in tissues, including the kidneys, lungs, liver, heart, tendons, gastro-intestinal tract, and spleen [110,111,112]. These animal studies, although very heterogeneous in design and with relatively short follow-up, show the involvement of dietary AGEs in the development of type 1 and type 2 diabetes [113]. In addition, animal studies show that high-AGE diets have far-reaching effects throughout the body, including adverse effects in the bones [114,115], the lungs [116], the vascular [117,118,119,120] and neurological systems, and on the natural aging process [113,121,122]. AGEs contribute to adverse health effects, predominantly through the induction of chronic inflammation (for review, see [123]).

Translation of these findings to humans is difficult, especially when assessing life-long intake and accumulation of AGEs. However, epidemiological studies have confirmed associations between dietary intake of AGEs and pathological conditions [124]. Although not specific to AGEs, a systematic review on nutrition and COPD demonstrated very strong evidence that processed meat elevates the risk of COPD development [125]. Negative associations between processed meat intake and lung function in general have also been shown [126]. Moreover, interventions with a diet low in AGEs (<15,000 kU) improve symptoms in patients with diabetes or chronic kidney disease and improve insulin sensitivity in healthy, overweight individuals [127,128].

Notably, approximately 80% of dietary AGEs cannot be readily absorbed but require digestion by the gut microbiota first. The intake of AGEs is associated with alterations in the gut microbiome because the gut microbiota can use AGEs for their own metabolism. A high AGE intake was, for instance, associated with reduced diversity and richness of the gut microbiome, with lower amounts and fewer strains of *Bacteroides*, *Bidifobacteria*, and *Lactobacilli* [129]. A diet high in AGEs furthermore downregulated expression of the junctional proteins occludin and claudin 1, which was associated with more circulating LPS in rats [130]. Therefore, high dietary AGE intake can also indirectly contribute to chronic inflammatory diseases by inducing microbial dysbiosis and increasing gut permeability. Similarly, gut microbial dysbiosis has been associated with an increased severity of COPD in patients treated with corticosteroids [131].

### 5.3. Enhanced AGE Formation through Inhalation of Cigarette Smoke

Smoking is an important risk factor for many chronic inflammatory diseases, including COPD, CVD, and osteoporosis. In 1997, Cerami et al. reasoned that because the curing of tobacco leaves takes place under AGE-producing conditions, it could constitute a source of AGE precursors. They went on to show that an aqueous extract of tobacco leaves, as well as cigarette smoke condensate, can indeed form AGEs in collagen in vitro. Further experiments demonstrated that tobacco as well as smoke condensate contain glycotoxins (i.e., reactive and unstable (di)carbonyl-containing glycation products) that react fast (within hours) with proteins to form AGEs. Examples of these glycotoxins include formaldehyde, acetaldehyde, acrolein, glyoxal, and methylglyoxal, which are known to be responsible for some of the toxic effects of smoking [132]. Enhanced accumulation of AGEs can be seen through increased autofluorescence in the skin of smokers compared to non-smokers [133,134]. Importantly, RAGE expression is also increased in the lung tissue of smokers and in animal models of smoke exposure, particularly in the bronchiolar epithelium, reactive pneumocytes, alveolar macrophages, and endothelial cells. Most epithelial RAGE staining was observed in close proximity to inflammatory cell infiltrates [83,135,136].

Glycotoxins can be easily absorbed through the lungs and react with serum proteins. Higher serum AGE levels and glycated ApoB have indeed been reported in smokers compared to non-smokers [132]. Increased AGE levels are also present in other organs of smokers, including blood vessels and lens tissue [137]. Moreover, it was shown that approximately half of the vascular injury induced by tobacco smoke exposure in isolated rat carotid arteries was attributed to the glycotoxin glyoxal [138]. AGEs are also increased in the urine of smokers compared to non-smokers [139].

The formation of glycotoxins starts during the curing and aging of tobacco leaves, and the addition of sugars to cigarettes to improve taste further augments glycotoxin levels and, therefore, the formation of AGEs in the lungs. Yet, although it was demonstrated that higher levels of acrolein and formaldehyde are formed from cigarettes to which sugars were added, no clear effects on dependence in humans or health effects in rodent inhalation models could be observed [140]. Moreover, e-cigarettes, especially fruit-flavored ones, also contain abundant levels of glycotoxins, which warrant careful long-term toxicological monitoring [141].

Although the composition of biomass smoke can vary depending on the fuel source and the combustion conditions, the major harmful components are generally similar and comparable to those of cigarette smoke [142]. It contains high concentrations of hazardous inhalable particulate matter and volatile chemicals, including aldehydes that can give rise to AGEs. A comparison of the glycotoxin content and production of AGEs has so far not been made.

### 5.4. Limitations in Physiological Detoxification

Because of the detrimental effects of AGEs, cells are equipped with a number of mechanisms that enable their detoxification and/or elimination (Figure 5). Importantly, phase 1 and 2 detoxification enzymes are not involved in this process. Instead, intracellular glycated proteins are degraded by the ubiquitin-proteasome system, or autophagy. Ligand-induced endocytosis through various receptors is the first step toward the elimination of extracellular AGEs. Scavenger receptors, as well as AGER1, AGER2, and AGER3, and Stab1 and Stab2, bind high-molecular-weight AGEs, which, after internalization, are degraded by the lysosomal system [143]. Importantly, the binding of AGEs to these receptors does not trigger intracellular signal transduction. Instead, it results in their lysosomal breakdown into low-molecular-weight AGEs, which enter the circulation and can be excreted through the urine. Diminished kidney function or overt kidney disease limits the excretion of AGEs and can contribute to their tissue accumulation and pathogenic effects [144]. AGE accumulation is also attributed to the bulky nature of glycated and aggregated proteins, which limits their entry into the proteasomal core. Lastly, the age-related decline in the protein quality control systems contributes to their accumulation.

Our body is also equipped with a system that prevents endogenous AGE formation, especially through the highly reactive dicarbonyls glyoxal and methylglyoxal. Glyoxalase enzymes detoxify these AGE precursors to D-lactate in a glutathione (GSH)-dependent manner. The major regulator of anti-oxidant gene expression, nuclear factor erythroid-derived 2 like 2 (Nrf2), controls the gene transcription of glyoxalase 1, which is the rate-limiting enzyme in the detoxification reactions. Because Nrf2 also controls many enzymes involved in maintaining GSH levels, it also influences glyoxalase 1 activity. As such, depleted GSH levels and/or defects in Nrf2 activation that are observed in chronic inflammatory diseases negatively regulate glyoxalase activity and can contribute to the enhanced formation of dicarbonyl-derived AGEs.

Reduced expression and/or activity of glyoxalase 1 have been reported in normal aging as well as in multiple chronic diseases, including diabetes, atherosclerosis, and hypertension [145]. In these diseases, increased levels of methylglyoxal and methylglyoxal-derived AGEs are found as well. In COPD, the glyoxalase system has not been examined. For a recent review of the glyoxalase system, see [146].

The accumulation of AGEs depends on the balance between endogenous formation and exogenous intake on the one hand and detoxification and elimination on the other (Figure 5). Importantly, all these aspects are deregulated in age-related chronic inflammatory disorders and result in enhanced local and systemic AGE levels.

## 6. Common Mechanisms Driven by AGE-RAGE

### 6.1. Local RAGE-Driven Inflammatory Signaling and Aging

Ligand-activated RAGE signaling promotes and sustains local and systemic inflammation. Although AGEs were the first ligands identified to bind RAGE, the capacity to induce RAGE signaling and inflammatory reactions varies between different AGEs. CML, CEL, and MGH-1-modified peptides bind the V domain of RAGE and induce intracellular signaling. However, other AGEs, such as pentosidine, do not induce any RAGE signaling [147,148]. A comprehensive investigation of the RAGE-activating capacity of the full spectrum of AGEs is urgently needed. Regardless, RAGE signaling promotes local and systemic inflammation. Given the positive feedback signaling of RAGE and its ligands, which are classically released upon tissue damage, RAGE is potentially a key mediator of inflammaging. Inflammaging is the sterile, non-infectious, low-grade inflammation that is associated with aging and is an important hallmark of age-related chronic inflammatory diseases. AGEs activate various inflammatory cells through RAGE, including monocytes, neutrophils [149], and mast cells [150]. RAGE activation in these local and resident immune cells causes the release of inflammatory mediators, more RAGE ligands, and increased RAGE expression. Interestingly, RAGE activation and AGE modification can repress bacterial killing and can inhibit the function of antibacterial proteins [151,152]. Studies have shown that RAGE also plays a role in acute infections, with divergent protective or pathogenic roles depending on the causative agents; however, the majority have shown that inhibition of RAGE is protective [153]. In addition to activating immune cells, AGEs and RAGE also facilitate their adhesion and migration into tissues [154,155].

Next to the activation of the innate immune system through RAGE signaling, glycated proteins can be presented by antigen-presenting cells to activate T-helper (Th) cells and trigger adaptive immune responses. RAGE expression on T-lymphocytes plays a role in this T-cell activation and differentiation towards Th1 [156]. The activation of T-cells by AGEs was shown to play a role in the pathogenesis of type 1 diabetes [157], as well as other autoimmune diseases [158,159].

AGE-RAGE-driven chronic inflammation can also promote processes that hinder regeneration and accelerate the natural aging process, including stem cell dysfunction. For instance, AGE-induced RAGE expression in mesenchymal stem cells resulted in cell death and diminished chondrogenic differentiation [160]. Endothelial progenitor function and angiogenesis are also impaired by AGEs and RAGE [161,162]. AGEs and RAGE signaling can also promote other hallmarks of aging, including increased genomic instability (through enhanced production of ROS and dysregulated DNA repair) and mitochondrial dysfunction (AGEs target various mitochondrial proteins and disrupt ATP production [163]).

AGEs and RAGE signaling are furthermore associated with fibrosis, for example through the induction of apoptosis, TGFβ signaling, epithelial-to-mesenchymal transition (EMT), and smooth muscle proliferation.

### 6.2. Local RAGE-Independent Effects of AGEs

In addition to inducing RAGE signaling, AGE-modified proteins display structural or conformational changes that can cause dysfunction. For instance, glycation of the 20S proteasome inhibits its activity, which is involved in the accumulation of damaged proteins [164]. Heat shock proteins 27 and 90 can be modified by methylglyoxal, which could play a role in aging [165]. Most of these modifications and their impact have been demonstrated in purified proteins or in cell culture experiments. The impact of these functional alterations due to AGE modifications in vivo remains to be determined. Their relevance is also questioned because these proteins are typically degraded and replaced by unmodified ones. In contrast, the age-related deregulation of proteostasis makes it more likely that it has an impact and can indeed contribute to disease pathogenesis.

Regardless, AGE modification is especially relevant for long-lived proteins. Intracellular long-lived proteins include proteins that form nuclear pore complexes, histones, crystallins, and myelin sheets. Consequences of AGE accumulation in these proteins include epigenetic alterations, aberrant gene expression, and neurological problems, most of which are common among chronic inflammatory disorders [166,167,168].

Long-lived proteins are also abundant in the ECM, where collagen has a half-life of up to 244 days, while the lifespan of elastin is equal to that of the human lifespan and is believed to limit the human lifespan. Skin autofluorescence reflects the life-long accumulation of AGEs in the ECM. AGE modification of ECM proteins has important functional consequences (summarized in Figure 6). This includes altering cell matrix interactions, protein crosslinking, and reducing sensitivity to degradation, leading to accumulation and stiffness of the matrix. Glycation of elastin, on the other hand, enhances its degradation, which likewise contributes to stiffening. Tissue stiffening is an important hallmark of normal organ aging, including of the skin, lungs, arteries, joints, cartilage, bones, heart, skeletal muscle, and lens. Furthermore, tissue stiffening is accelerated in age-related diseases, including COPD, hypertension, and atherosclerosis. Interestingly, the accumulation of intra- and intermolecular covalent bonds between elastin and collagen in the ECM, amongst others, is induced and facilitated by glycation and forms the basis for the cross-linking or glycation theory of aging, postulated by Johan Bjorksten in 1942 [169]. Glycation of ECM proteins promotes aberrant repair and tissue fibrosis through defective matrix turnover, as well as the migration, proliferation, and differentiation of fibroblasts into myofibroblasts [170,171]. Smooth muscle cells are similarly triggered to proliferate and become activated by AGE-modified ECM [172]. Altered cellular functioning through mechanosensing induced by glycated ECM is involved in these outcomes.

Basement membrane thickening is another ECM-related feature of normal aging that can be observed in capillaries, kidneys, lenses, and lungs. AGE modification of basement membrane ECM proteins, such as collagen IV, can cause thickening. In addition, AGE modification of the basement membrane triggers RAGE signaling in epithelial and endothelial cells, which can cause reduced endothelial and epithelial cell adhesion and angiogenesis [173,174]. Furthermore, AGE modifications of the basement membrane are associated with leakiness of the gut, the blood-brain barrier, and the blood-retina barrier. Leakiness of such barriers contributes to systemic inflammation, which can promote disease progression at other sites of the body.

The glycation of proteins, especially long-lived proteins, has significant long-term effects on the processes of aging and the progression of age-related diseases. These protein modifications contribute to protein dysfunction, matrix accumulation, and disruption of barrier sites, which ultimately lead to tissue stiffening and a reduction in organ function. This is relevant to the progression of COPD as well as many of its comorbidities, most notably the cardiovascular system.

### 6.3. Systemic Effects—Interorgan Crosstalk

The described mechanisms that lead to enhanced AGE formation and accumulation in tissues also lead to enhanced AGE levels in the circulation. In turn, these circulating AGEs may exacerbate the negative effects of AGE-RAGE signaling in various organs and may contribute to the development and progression of COPD and its co-morbidities.

Circulating AGEs include free AGEs as well as glycated plasma proteins such as albumin, LDL, and HDL (Figure 7). Circulating AGEs trigger RAGE activation on endothelial cells, circulating immune cells, and platelets (Figure 7 and Figure 8). Importantly, glycation potentiates the pathogenic role of LDL and attenuates the protective role of HDL. For glycated LDL, it is shown that its uptake by lipoprotein and scavenger receptors is limited [175], and instead it is phagocytosed by macrophages to form foam cells [176,177]. Glycated HDL and LDL are more sensitive to oxidation and also induce more ROS production compared to their non-glycated counterparts [178]. In addition, the protective effects of HDL, including the induction of antioxidant defenses, are attenuated by glycation [179,180]. In vitro studies with glycated albumin show it can induce inflammation in macrophages and trigger insulin resistance in skeletal muscle cells [181,182]. Moreover, membrane proteins in red blood cells (CD233, bands 3 and 4.1) and spectrin can be glycated, which increases their adhesion [52]. Red blood cell glycation is increased in patients with diabetes and is associated with their phagocytosis by endothelial cells. It constitutes an important source of RAGE signaling in endothelial cells, leading to inflammatory signaling and ROS production through NADPH oxidases [183,184]. Furthermore, glycation of red blood cells limits their deformability [185], which can contribute to damage to the microvasculature as seen in diabetes. Of note, AGEs are formed during the storage of red blood cells, which can cause transfusion damage, including in the lung microvasculature [186,187]. Within red blood cells, glycated hemoglobin has a higher oxygen affinity compared to non-glycated hemoglobin, which can play a role in the reduced exercise capacity of diabetics [188].

These circulating AGEs activate endothelial cells throughout the body, which leads to oxidative stress, inflammation, inhibition of eNOS activity, and increased vascular permeability [189]. A close correlation is found between circulating AGE levels and RAGE expression in endothelial cells [190], as well as between serum AGEs and sRAGE levels [191], indicating endothelial activation by these circulating ligands. In monocytes, AGEs stimulate the production of IL-6, which triggers CRP release by the liver. CRP consequently upregulates RAGE expression in macrophages and endothelial cells, which exacerbates inflammation [192]. AGEs also strongly activate platelets and trigger their aggregation, which is associated with increased expression of RAGE and reduced eNOS activity [193,194]. Platelets are pivotal players in many inflammatory diseases because chronic inflammation induces platelet hyperreactivity, which may in turn accelerate inflammation [195]. AGE-RAGE signaling appears to play an important role in this amplification cascade.

There is a close association between AGEs and lipid metabolism and obesity. High fat diets in humans, as well as in animal models generate high levels of AGEs, as well as other RAGE ligands [196,197]. Moreover, RAGE expression is increased in macrophages and adipocytes of obese subjects [198]. On the other hand, *AGER* knockout mice are protected from high fat-induced weight gain, and have higher glucose and insulin tolerance. In addition, they not only have lower numbers of macrophages in their visceral adipose tissue deposits, but these macrophages also displayed more M2 or anti-inflammatory phenotypic features. It was furthermore demonstrated that these protective effects were mostly due to RAGE expression in myeloid cells [199]. Obesity is also linked to the glyoxalase system, with genetic studies in human and mice demonstrating associations between body weight/BMI and glyoxalase 1 expression [200,201]. A functional role of glyoxalase 1 in the development of obesity is furthermore implicated from a conference report that describes that glyoxalase 1 overexpressing mice are protected from diet-induced weight gain [202]. The AGE-RAGE signaling could therefore constitute a common pathogenic mechanism, particularly in overweight or obese patients.

## 7. The AGE-RAGE Axis in Age-Related Chronic Inflammatory Diseases

### 7.1. AGE-RAGE in COPD

Evidence from clinical studies indicates that the expression of RAGE and the accumulation of AGEs are increased in patients with COPD. Studies showed increased RAGE expression using immunohistochemistry in the submucosa, epithelium, and smooth layers of bronchial walls [203], and in the alveolar macrophages [136] and alveolar walls [204] of COPD patients compared to non-smokers and smoking controls. Increased RAGE expression in whole lung lysates was reported in COPD [205]. In contrast, lower sRAGE levels were observed in the bronchoalveolar lavage fluid (BALF) of COPD patients [206]. Increased AGE staining was also prominent in macrophages, alveolar, and airway walls of COPD patients [204], while CML levels were increased in the peripheral airway epithelial lining fluid of COPD patients [207]. This latter study also showed a positive association between CML levels and the inflammatory chemokine IL-8. However, in another study, these AGE/(s)RAGE alterations were not present in bronchial biopsies and sputum of COPD patients [208]. This discrepancy could have been due to differences in the regions of the lungs examined and the methods employed for collection and analysis.

The role of RAGE signaling in the development of COPD is demonstrated using mice in which RAGE levels are genetically manipulated. Conditional overexpression of *AGER* in adult murine lungs increased air space enlargement and inflammation similar to COPD [209], whereas conditional *AGER* knockout mice were protected from cigarette smoke-induced neutrophil recruitment and emphysema [74,210]. Another study showed that after chronic cigarette smoke exposure, *AGER* knockout mice had reduced changes in respiratory mechanics and an altered immune response to acute cigarette smoke exposure [76]. The upregulation of RAGE expression was recapitulated in the elastase model of emphysema, and the RAGE inhibitor FPS-ZM1 could prevent inflammation and emphysema development [205]. Moreover, it was demonstrated that RAGE expression in structural cells but not in hematopoietic cells drives elastase-induced emphysema [211]. *AGER* knockout mice were also found to be protected against inflammation and airway hyperresponsiveness induced by acute smoking [212]. Similarly, Sanders et al. showed that cytokine production by *AGER*-deficient macrophages was attenuated in response to cigarette smoke extract and emphasized the role of RAGE on macrophages in generating oxidative stress [210]. The effects of AGEs and RAGE on lung tissue are summarized in Figure 8.

To date, no data exist on the contribution of RAGE to the cytotoxic effects of wood smoke exposure, but particulate matter in general has been shown to elicit effects in the lung through RAGE signaling. Chronic exposure to World Trade Center particulate matter, for instance, increased RAGE expression in the lungs of mice, while *AGER* knockout mice were protected from increased airway resistance, airspace enlargement, and lung tissue remodeling [213]. Diesel particulate matter also stimulates RAGE expression, which is involved in the release of pro-inflammatory mediators from lung epithelial cells [214]. However, it remains to be determined if RAGE signaling is stimulated by AGE moieties present on combusted diesel particles or if oxidation products, including AGEs, are formed during the combustion of diesel, as is seen in the burning of tobacco.

In addition to the mechanistic studies of AGE-RAGE alterations in animal models of COPD, several cross-sectional studies have consistently shown reduced systemic levels of total sRAGE in humans with COPD [206,208,215,216,217,218,219,220]. These studies also demonstrate a positive correlation between systemic sRAGE levels and lung function. Iwamoto et al. only found decreased plasma sRAGE levels in COPD patients compared to non-smokers and showed that a lower baseline sRAGE level could predict a more rapid lung function decline in smokers with COPD over a four-year follow-up period [220]. Smith et al. showed decreased plasma sRAGE levels in a small group of COPD patients, which further decreased during an acute exacerbation [215]. Furthermore, this was the only study that reported an association between sRAGE and inflammation as determined by CRP levels. Plasma sRAGE levels were further decreased in COPD patients with oxygen treatment compared to those without [218]. Another study found that lung and systemic levels of sRAGE were deficient in both subjects with asthma and COPD with neutrophilic inflammation compared to those without neutrophilia [206]. Importantly, we found plasma levels of both sRAGE and esRAGE to be decreased in COPD patients, but only sRAGE levels were associated with disease severity [217].

In general, decreased plasma sRAGE levels in COPD are associated with lung function loss but also, in particular, with the presence, severity, and even progression of emphysema [216,221]. This is consistent with the loss of mRAGE-expressing ATI cells in the progression of emphysema. Alternatively, lower plasma sRAGE levels in COPD could be due to altered expression of RAGE in the lungs or other organs or to altered shedding. Studies examining both compartments simultaneously are needed to address these possible mechanisms. Furthermore, decreased plasma sRAGE in COPD is associated with SNPs in the *AGER* gene, including G82S, which is also associated with lung function decline and the presence of emphysema. An additional SNP in the flanking region of *AGER* (rs20171278) was associated with decreased *AGER* gene expression in lung tissue and sputum sRAGE levels [208]. For a recent overview of the polymorphic variants in *AGER* related to COPD, we refer the reader to [222].

The evidence from animal models as well as clinical data indicates a prominent role for RAGE in the development of emphysema. It remains to be determined to what extent RAGE is involved in the development of the chronic bronchitis phenotype of COPD, which exhibits mucus accumulation in the airways. Unfortunately, murine cigarette smoke exposure models of COPD do not exhibit mucus hyperplasia, which limits the ability to examine the involvement of RAGE in this phenotype. Importantly, inducible RAGE transgenic mice overexpressing RAGE in a CCSP-dependent manner do not show signs of mucus metaplasia [209]. On the other hand, RAGE has been linked to mucus hypersecretion in various mouse models of allergic airway disease. For instance, RAGE knockout mice are protected from recombinant type 2 cytokine-induced airway mucus metaplasia, and treatment with a RAGE antagonist inhibits the induction of mucus in mouse lungs as well as primary human airway epithelial cultures [223]. In addition, adenoviral-mediated overexpression of sRAGE suppressed mucus production in a mouse model of neutrophilic asthma [224]. Mechanistically, RAGE was shown to be critically involved at multiple points in the process of developing allergic airway disease, including allergen-induced alarmin release, direct alarmin signaling, and downstream type 2 cytokine signal transduction, all of which culminate in the development of asthma-like features, including airway resistance and mucus metaplasia (reviewed in [225]). Moreover, various S100 proteins were shown to induce MUC5AC expression in lung epithelial cells. However, RAGE was only shown to play a role in the induction by S100A12 [226].

Studies examining changes in circulating AGE levels in COPD are scarce. Two studies found no difference in circulating CML levels in COPD patients compared to controls as determined by ELISA. There was also no association between CML and sRAGE or with inflammatory markers [216,218]. However, using more sensitive and reliable HPLC mass spectrometry, we did find increased levels of CEL and decreased levels of CML, but no changes in the levels of pentosidine in the plasma of COPD patients compared to controls [133]. Furthermore, we showed an association of CML and CEL with lung function but not with CRP. Another study that measured these same three AGEs in the plasma of COPD patients by ELISA only found elevated CML levels [208]. We and others also demonstrated enhanced AGE accumulation in the skin, measured as skin autofluorescence, in patients with COPD compared to both smoking and non-smoking controls [133,134,208,227].

Recently, we identified four clusters of COPD patients based on systemic markers of inflammation. Pathway analyses identified the enrichment of the RAGE pathway in three of these clusters, suggesting that there is indeed a role for RAGE not only in lung pathology but also in systemic manifestations of COPD [228]. Overall, data from animal models and human studies suggest that the AGE-RAGE signaling axis contributes to the development and progression of COPD.

### 7.2. AGE-RAGE in Insulin Resistance, Diabetes, and Diabetic Complications Diabetes

AGEs have been most thoroughly studied in diabetes, where circulating levels of various AGEs are increased [229,230,231,232,233]. It is thought that this results from accelerated AGE formation due to the presence of oxidative stress and hyperglycemia. However, AGEs are not merely a result of the disease, as circulating AGE levels are also associated with insulin resistance in pre-diabetes [234,235]. AGEs have indeed been shown to contribute to β-cell dysfunction and insulin resistance. RAGE expression has been observed in pancreatic islet cells, and RAGE signaling has been shown to be involved in mediating the negative effects of AGEs on β−cell function (Figure 8) [236,237]. Furthermore, esRAGE co-localizes with insulin in secretory granules, but it is not known if this has any physiological function [238]. Importantly, levels of circulating AGEs and enhanced skin autofluorescence are associated with diabetic complications (reviewed in [239,240,241,242,243]). AGE accumulation is also seen at sites affected by diabetic complications, such as the eyes [229] and kidneys [244,245,246,247]. Notably, the expression of RAGE is also increased at sites of diabetic complications, including the kidneys [247] and atherosclerotic plaques [248]. Circulating and skin AGE levels are therefore examined as potential biomarkers for diabetic complications (Figure 8).

Surprisingly, sRAGE and esRAGE are divergently altered in the plasma of diabetics and have opposing associations with inflammation and CVD. Plasma levels of sRAGE are mostly found to be increased in patients with diabetes compared to controls [191,249,250,251,252,253] and are positively associated with inflammatory markers [254,255,256]. Increased sRAGE is also a prognostic marker for coronary artery disease and cardiovascular events in patients with diabetes [191,257]. Moreover, various polymorphisms in the *AGER* gene have been linked to diabetes, its complications, and plasma sRAGE levels [258].

Conversely, plasma esRAGE levels are decreased in diabetes and pre-diabetes patients [251], with a negative correlation with inflammatory markers [259,260,261]. Negative correlations of plasma esRAGE levels with markers of macrovascular complications have also been reported [259,260,261,262,263,264].

Data regarding esRAGE levels are consistent with the protective and anti-inflammatory effects of soluble RAGE, whereas sRAGE data are in contrast to the postulated function as a negative regulator of RAGE-induced inflammation. These studies suggest that mechanisms of RAGE shedding and alternative splicing are divergently affected in diabetes. However, studies that examine both at the same time have not been executed. Importantly, esRAGE is predominantly expressed in endothelial cells, yet the main sources of circulating esRAGE and sRAGE remain the same in physiological and disease conditions. Lastly, studies examining the specific roles of sRAGE and esRAGE in the circulation with respect to inflammation and cardiovascular effects are needed.

Nonetheless, the data collectively suggest that the AGE-RAGE axis has a pathogenic role in the development of diabetes as well as typical diabetic complications. In addition, studies have linked diabetes to cognitive impairment, an increased risk of osteoporosis, and subclinically attenuated lung function [265,266,267]. However, the potential mechanistic role of AGE-RAGE signaling in these less well-established complications remains to be determined. By extension, the AGE-RAGE axis may play a role in the development of these diseases independent of diabetes and its complications, as other mechanisms can promote AGE accumulation.

The persistent threat of hyperglycemia in uncontrolled diabetes provides favorable conditions for enhanced AGE formation in the circulation, which may also manifest in the lungs, where it may exacerbate the negative effects of AGE-RAGE signaling in COPD. Interestingly, levels of circulating sRAGE appear to be increased in diabetes, whereas they are decreased in COPD. It is possible that this could influence the use of sRAGE as a biomarker of lung function decline in COPD patients with diabetes. Nonetheless, further research is needed to examine the potential contribution of AGEs produced in diabetic conditions to the development and progression of COPD in those who are co-morbid.

### 7.3. AGE-RAGE in Cardiovascular Diseases

Increases in RAGE protein levels are seen with increasing age in the heart muscle and in vascular tissue. In particular, enhanced RAGE expression in the arteries has been shown to be a dominant risk factor for CVD [268,269]. Similarly, with increasing age, CML deposits are found in the elastic membrane and intimal extracellular matrix of arterial walls [270]. Additional deposition of AGEs is found in the media and adventitia of atherosclerotic vessels, with rupture-prone carotid atherosclerotic plaques appearing to accumulate even more CML and MGH-1 [271]. Foamy macrophages also stain positive for AGEs [270,272]. After a myocardial infarction, enhanced CML levels were found in intramyocardial blood vessels [273]. In patients with atrial fibrillation, more CML was present in vessels of the left atrial myocardium and in fat tissue, which correlated with inflammation [274].

Mechanistic studies in animals support the role of RAGE and AGEs in the development of cardiovascular pathology; the mechanisms are summarized in Figure 8. *AGER* knockout animals are protected from developing atherosclerosis in both non-diabetic and diabetic animal models [275]. Similarly, exogenous sRAGE administration protected mice from developing atherosclerosis [58,276,277,278]. Furthermore, AGE inhibitors were able to prevent the calcification of atherosclerotic lesions, while a RAGE agonist induced calcifications ex vivo [279]. RAGE inhibition through sRAGE also protected mice against systolic overload-induced heart failure [280].

In addition to the enhanced accumulation of AGEs in heart tissue and vasculature in acute and chronic CVDs, more AGEs are also found in the circulation. The Baltimore Longitudinal Study of Aging and the Health ABC study reported increased serum levels of CML in healthy older adults, which were associated with arterial stiffness [281,282]. Hypertensive patients also display increased serum AGE levels compared to normotensive controls [283], and circulating AGE levels correlate with the degree of atherosclerosis [284,285]. Pentosidine levels in serum are also increased in patients with atrial fibrillation [286] and are found to be an independent prognostic factor in heart failure [287].

With respect to circulating sRAGE levels, divergent findings are reported in CVD. Patients with coronary artery disease [288,289] and essential hypertension [290] had reduced circulating sRAGE levels. Lower circulating sRAGE levels were independently associated with increased arterial stiffness [291] and the future development of heart failure [292]. In addition, plasma esRAGE levels were inversely associated with atherosclerosis [259].

Conversely, another study found that patients with high levels of plasma sRAGE were shown to be at increased risk for future cardiovascular events [293]. In acute settings, such as in patients experiencing acute myocardial infarction [294], myocarditis [295], or atrial fibrillation [286], circulating sRAGE levels were shown to be significantly increased.

Overall, attenuated sRAGE levels appear to relate to chronic CVDs, whereas more acute cardiovascular events present with increased sRAGE levels. These divergent effects in acute versus chronic disease settings are also seen in acute versus chronic lung pathologies, such as ARDS versus COPD. Studies are needed to examine the potential causes of the changes in circulating sRAGE levels and their potential contribution to disease. This is especially relevant in the context of exploring sRAGE as a disease biomarker of lung function decline in COPD patients with and without comorbidities. Importantly, the enhanced circulating levels of AGEs reported in both COPD and CVD likely exacerbate the negative effects of AGE-RAGE through endothelial cells and cells in the circulation, especially in the context of attenuated circulating sRAGE levels (Figure 7).

Another important observation is that cardiovascular mediation with anti-inflammatory properties affects RAGE. Statins, for instance, inhibit RAGE expression in atherosclerotic lesions [296], and patients with hypercholesterolemia treated with statins demonstrate higher plasma sRAGE levels [297]. Increased sRAGE levels are also observed in patients treated with an ACE inhibitor [298]. The use of angiotensin II receptor blockers also attenuated the expression of RAGE and was associated with lower sRAGE levels [299]. It is not known to what extent the anti-inflammatory properties of these drugs are RAGE-mediated. The anti-inflammatory effects of corticosteroids could be exerted through RAGE, but at this point, there is no evidence to support this hypothesis. Similarly, in COPD, no association of plasma sRAGE levels with corticosteroid use has been reported. Regardless, these data illustrate that the use of medication should always be taken into account as a possible confounder when investigating the AGE-RAGE axis in chronic inflammatory diseases.

### 7.4. AGE-RAGE in Osteoporosis

AGEs are expected to accumulate in bone due to the low turnover of bone tissue and the high content of long-lived proteins, most notably collagen 1. During normal, healthy aging, the pentosidine content of the cortical and trabecular bones does increase exponentially [300], which is negatively associated with bone density and mineralization [301]. Furthermore, bones from osteoporosis patients exhibit increased CML and imidazolone immunoreactivity, which are negatively correlated with the percentage of bone surface covered with osteoblasts [302]. Although osteoblasts, chondrocytes, and osteoclasts express RAGE, there is currently no data on RAGE expression in bone tissue or on circulating sRAGE levels in osteoporosis [303,304].

Mice lacking RAGE display increased bone mass and bone mineral density in combination with decreased bone resorption, indicating a role for RAGE in osteoclast maturation and function [305,306,307]. This is consistent with the role of RAGE in monocyte maturation into macrophages, with which osteoclasts share a common progenitor. Furthermore, RAGE-dependent anti-osteogenic effects of AGEs have been demonstrated, as RAGE activation was found to inhibit osteoblast proliferation by suppressing WNT, ERK, and PI3K signaling [308]. These studies indicate that AGE-RAGE signaling does indeed contribute to increased bone turnover and the development of osteoporosis (Figure 8).

Increased circulatory AGE levels have also been reported in osteoporosis patients, which are negatively correlated with bone mineral density [309,310]. Elevated circulating AGE levels can be due to enhanced bone resorption, which releases AGEs from the bone matrix. Alternatively, AGEs could be part of a mechanistic pathway that is actively involved in bone turnover. In support of this, serum pentosidine levels are a determinant of vertebral fractures in postmenopausal diabetics, which is independent of bone mineral density [311,312]. In addition, increased urinary pentosidine levels were an independent risk factor for osteoporosis-induced vertebral bone fractures [313], and the performance of the fracture and immobilization score (FRISC) tool to predict fractures was improved when urinary pentosidine values were added [314]. Lastly, circulating pentosidine levels were significantly related to cellular parameters of bone turnover in osteoporosis patients [310]. It is important to note that anti-resorptive drugs decrease the removal of AGEs accumulated in the bone matrix, which potentially negatively impacts osteoporosis [315].

While available data regarding RAGE expression and sRAGE levels in osteoporosis indicate that AGE levels are increased in the bone and circulation in osteoporosis, AGE levels also seem to strongly predict prognosis. In addition, similar to other comorbidities, elevated circulating levels of AGEs associated with osteoporosis may contribute to the progression of COPD and other affected comorbidities.

### 7.5. AGE-RAGE in Sarcopenia

RAGE signaling plays a role in the development of skeletal muscle by determining the fate of myogenic progenitor cells. It is similarly involved in satellite cell reactivation in adult life, being re-expressed after muscle injury. Here, it plays divergent roles in settings of acute versus chronic muscle inflammation. In acute inflammation, it promotes myoblast proliferation and differentiation, with *AGER* knockout mice displaying delayed regeneration after injury [316]. On the other hand, RAGE activation not only amplifies chronic inflammation, but it also induces myocyte apoptosis and the loss of satellite cells, thereby potentially contributing to sarcopenia and cachexia [317]. Other animal studies further support the role of AGE-RAGE in the development of muscle loss with aging and in disease settings (Figure 8). Mice fed a diet rich in AGEs for 16 weeks showed CML accumulation in muscle tissue, which was associated with reduced muscle mass and strength [115]. Pharmacological inhibition of RAGE ameliorated aging-induced loss of muscle mass in mice [318]. Moreover, AGE accumulation and RAGE expression were increased in the muscle tissue of mice subjected to the hind-limb suspension model of atrophy. In addition, administration of a RAGE antagonist reduced muscle atrophy and the expression of inflammatory markers [319].

In humans, the pentosidine content of skeletal muscle in older subjects is increased compared to younger adults [320]. Currently, there are no data available on AGE accumulation in the muscle of subjects with sarcopenia. However, numerous studies have shown skin autofluorescence and systemic AGE levels are positively correlated with loss of muscle mass and strength in several different populations, most notably the elderly [321,322,323,324,325,326,327]. Circulating AGE levels are also associated with frailty [328,329].

On the other hand, data regarding circulating sRAGE levels are inconsistent. One cross-sectional Korean study found an association between low levels of sRAGE and low muscle mass [330]. However, other cohort-based studies reported increases in plasma sRAGE in individuals with sarcopenia and suggested sRAGE as a biomarker for frailty and mortality in frail individuals [331,332,333]. Differences in the study design, including sex-associated differences, may have contributed to these discrepancies [331].

In aggregate, data suggest that the AGE-RAGE axis contributes to loss of muscle mass and strength with aging and in disease settings through multiple mechanisms. This includes increased stiffening of muscle extracellular matrix, altered behavior of satellite cells [334,335], the induction of inflammatory signaling and microvascular damage, as well as effects on enzymes involved in cellular energy metabolism [336]. Moreover, elevated circulating levels of AGEs may contribute to the progression of COPD and other affected comorbidities, especially when plasma sRAGE levels are attenuated.

### 7.6. AGE-RAGE in Renal Disease

In the kidneys, low basal RAGE expression is present only in podocytes, but it is upregulated in nephropathy [246]. In addition, renal endothelial cells express RAGE, which plays a role in the induction of renal vascular injury [337]. Decreased kidney function attenuates the clearance of free AGEs from plasma. This leads to the trapping of plasma AGEs in the kidneys, which contributes to local accumulation and toxicity (Figure 8). Increased formation of AGEs also contributes to increased AGE levels in chronic kidney disease (CKD) [338]. AGEs accumulate primarily in the basement membranes of the kidneys, which is associated with the upregulation of RAGE in podocytes [246,339].

Reduced clearance by the kidneys is also expected to increase circulating AGE levels in patients with CKD. Marked increases in circulating AGEs, with up to a 40-fold increase for free AGEs, are indeed reported in patients on hemodialysis [340]. Importantly, serum CML levels are negatively associated with the estimated glomerular filtration rate (eGFR), a measure of kidney function [341]. Blood monocytes from CKD patients show enhanced RAGE expression, which is negatively associated with eGFR [342,343]. Several studies have suggested that high plasma AGE levels and skin autofluorescence are biomarkers for CKD progression and mortality risk [344,345,346,347,348]. Moreover, the glycation of the plasma proteins albumin and β2-microglobulin causes abnormal refolding, leading to structural characteristics resembling amyloid fibrils, which are prone to aggregate. Therefore, glycated albumin and β2-microglobulin could potentially be involved in the pathogenesis of dialysis-related amyloidosis [349,350]. Decreased kidney function also attenuates the clearance of sRAGE from plasma, which is increased in CKD patients [351,352]. Both esRAGE and sRAGE were also independently associated with all-cause mortality [353].

In animal models of diabetes, mice overexpressing *AGER* display increased glomerular hypertrophy, albuminuria, serum creatinine, and advanced glomerulosclerosis [354], whereas *AGER* knockout mice are protected from these pathogenic features [355]. *AGER* knockout mice are also protected from age-related renal lesions [356]. Systemic injection of AGEs, on the other hand, promoted the thickening of the basement membrane and expansion of the mesangial layer [357]. These studies suggest that AGE-RAGE plays a role in vascular damage in the kidneys, the development of glomerulosclerosis, and the activation of podocytes, which are all features of CKD [358].

Kidney function should furthermore be considered an important determinant of circulating levels of AGEs and sRAGE, and renal disease can thereby influence the development and progression of COPD and comorbidities.

### 7.7. AGE-RAGE in Depression and Anxiety

Patients with chronic inflammatory diseases often present with depressive symptoms, and chronic low-grade inflammation is thought to be involved in the development of these symptoms [359]. Although this could implicate AGE-RAGE signaling, only a limited number of studies to date have examined its involvement. In type 2 diabetics, being a young female of low socioeconomic status and education level, having a high BMI, and having high HbA1c levels were found to be risk factors for depression [360]. In another study, high skin autofluorescence was found to be independently associated with depressive symptoms and the presence of depressive disorder, whereas no such associations were found with plasma levels of AGEs [361]. Depression in type 2 diabetes is also negatively correlated with esRAGE levels in a Chinese population [362]. Lastly, low serum sRAGE levels were correlated with overt mental illness, including anxiety and severe depression [363].

Given the impact and relevance of anxiety and depression, further studies are warranted that examine the involvement of AGE-RAGE signaling. These studies should also examine the contribution of AGE-RAGE signaling to anxiety and depression as comorbidities in COPD.

## 8. Therapeutic Implications

If AGEs and RAGE signaling constitute common pathogenic mechanisms in COPD and associated chronic inflammatory diseases, single therapeutic strategies could collectively alleviate patients’ multimorbid conditions. From a prevention perspective, lifestyle interventions should be most effective. These lifestyle interventions can also limit any further accumulation of AGEs, reduce RAGE signaling, and thereby slow the progression of diseases. These interventions include refraining from smoking or quitting smoking. Dietary advice should be focused on limiting the intake of AGEs, which can include alternative preparation of food to avoid AGE formation during cooking and consumption of processed food. Glucose and fructose intakes, as well as the intake of saturated fats, should also be limited. Importantly, breast-feeding exposes infants to 70% fewer AGEs compared to formula feeding [364], a beneficial effect in addition to the other well-known advantages of breast-feeding. Similarly, limiting AGE accumulation is one mechanism that contributes to improved health through exercise. An exercise program, or its combination with an energy-restricted diet and the obesity drug orlistat, does decrease serum and urinary AGE levels [365,366]. In aged rats, a moderate-intensity exercise treatment reduced AGE levels and improved endothelial-derived relaxation, pulse wave velocity, and markers of oxidative stress [367].

When looking into pharmacological interventions (Figure 9 and Table 1), a couple of different compounds that prevent AGE formation have been tested, mainly in patients with diabetes. Aminoguanidine is, for instance, a scavenger of reactive dicarbonyls that constitute important drivers of AGE formation. In randomized clinical trials, it was found to prevent the progression of retinopathy in patients with diabetes. Although these studies demonstrated that AGEs do contribute to diabetic complications and were promising, serious side effects terminated further research into aminoguanidine [368]. The vitamin B6 metabolite pyridoxamine also functions as a dicarbonyl trap and was shown in phase 2 studies in diabetic nephropathy to slow disease progression. This was associated with reduced urinary secretion of TGFβ and attenuated the increase in plasma AGE levels [369]. The clinical trials examining the efficacy of pyridoxamine (PYRIDORIN) have unfortunately been halted due to financial issues. A number of other compounds also have AGE-lowering properties. The flavonoid quercitin, for instance, can trap glyoxal and methylglyoxal and thereby inhibit the formation of AGEs [370]. Metformin has also been shown to lower AGE production through improved glycemic control. The PPARγ agonist rosiglitazone also decreased AGE levels and increased serum levels of sRAGE [371]. Because these compounds have a wide range of actions, it is not clear yet to what extent their AGE-lowering properties contribute to their beneficial effects.

Another approach to limiting AGE formation could be to increase the expression and activity of glyoxalase 1. This could be accomplished through generic inducers of Nrf2, which include dietary components such as curcumin, resveratrol, quercitin, and sulforaphane. Of interest to note is that in vitro treatment of macrophages isolated from COPD patients with sulforaphane was able to restore the impaired phagocytosis of bacteria, which could potentially involve increased glyoxalase 1 expression [372]. Not all compounds, however, equally affect the expression of the approximately 890 Nrf2-regulated genes [373]. The particular combination of *trans*-resveratrol and hesperetin was found to potently induce glyoxalase 1 and reduce plasma levels of methylglyoxal and derived-protein glycation. This was associated with improved insulin sensitivity and improved arterial dilatation, as well as vascular inflammation, in overweight and obese participants in a phase 2A clinical trial [374]. These beneficial effects are likely the result of increased expression of many genes, not solely glyoxalase 1. Removal of free AGEs could alternatively be accomplished by enhanced expression of galectin-3/AGER3.

Alternatively, targeting RAGE directly is another potential therapeutic intervention strategy, as several small molecules that inhibit RAGE signaling have been developed, most notably FPS-ZM1 and Azeliragon (TTP488), both of which target the ligand-binding domain of RAGE, as well as inhibitors that target intracellular adaptor proteins that mitigate downstream signaling. Treatment with FPS-ZM1 was originally identified as a potential therapeutic molecule to treat Alzheimer’s disease but has also been shown to reduce AGE levels and improve outcomes in a variety of animal models of disease [367,375]. Furthermore, treatment with FPS-ZM1 alleviated elastase-induced emphysema in mice and reduced cigarette smoke-induced oxidative stress in vitro [205,376]. Azeliragon is an orally available small molecule that was found to reduce Aβ plaques and inflammatory cytokines while slowing cognitive decline in a preclinical model of Alzheimer’s disease [377]. Azeliragon was also tested in Phase 2b and Phase 3 clinical trials to treat mild-moderate Alzheimer’s disease. It was relatively well tolerated and safe in subjects; however, it failed to show efficacy [377,378,379]. On the other hand, a small molecule targeting the interaction of RAGE and DIAPH (RAGE229) was found to reduce inflammation as well as short- and long-term complications in a preclinical model of diabetes in mice [380,381]. In addition, it was recently discovered that RAGE also transduces signaling through SLP76 and that administration of a blocking peptide (TAT-SAM) reduced inflammation, tissue damage, and lethality in a mouse model of sepsis [382]. Further research is needed to determine the therapeutic potential of these RAGE-targeting strategies in preclinical models of COPD and multimorbidity and their potential use in humans, especially that of Azerliragon, which has some reported safety in humans already.

However, with any targeted therapy, it is important to note the potential negative impacts. The pathological roles of RAGE are many and continue to be identified. Conversely, while RAGE has proven important for development, its physiological role after development remains poorly understood. For instance, *AGER* knockout mice are viable even though it has been shown that natural RAGE expression is critical for normal development in mice. However, some abnormalities have been identified. *AGER* knockout mice exhibit subtle alterations in airspace structure as well as baseline lung compliance and elastance in aged mice [76]. In addition, mice lacking *AGER* spontaneously develop features of fibrosis in old mice [47,77]. However, it remains to be proven that these effects are directly due to the lack of RAGE expression and may be due to other alterations that are present in *AGER* knockout mice. For instance, *AGER* knockout mice overexpress ALCAM, while ALCAM-null mice overexpress RAGE. This may represent endogenous compensatory mechanisms, as both molecules are considered to be potential structural and functional homologs [383]. Nonetheless, abnormalities linked to *AGER* knockout mice are due to a global germline deletion of the *AGER* gene. Therefore, future studies are needed to determine any potential negative effects of exogenously blocking RAGE signaling in long-term settings, as many studies suggest targeting RAGE may represent a very promising therapeutic approach for many diseases.

Although the administration of sRAGE has been proven to be an effective therapeutic strategy in many animal models [354,358,384,385], clinical application is not a viable option since it will be difficult to produce this relatively large recombinant protein and to include the posttranslational modifications that are essential to its functioning. Moreover, as well as blocking RAGE signaling, sRAGE might limit beneficial interactions of RAGE with ligands involved in physiological responses, which are still poorly understood and might lead to unwanted detrimental effects. On the other hand, not all disease conditions described in this review display attenuated sRAGE levels. Upregulation of esRAGE could be a better approach because, for instance, it was able to restore the angiogenic response in diabetic mice [386].

All these options can only limit further AGE accumulation or RAGE signaling. Ideally, AGE-induced crosslinks should be reversed. This can be accomplished through so-called AGE breakers, of which Alagebrium (ALT-711, 4,5-dimethyl-3-phenacylthiozolim chloride) is the only compound tested in a clinical trial to date. Alagebrium reduced AGE accumulation and atherosclerotic plaques and provided protection against diabetic kidney disease and age-related myocardial stiffening in various animal models [387,388,389]. The only clinical study so far showed improved arterial compliance and lowered pulse pressure in older individuals after an 8-week intervention [390]. Unfortunately, no further clinical trials have been performed.

These interventions are and will first be tested in relatively simple animal models of single diseases—the combination of diabetes and complications at most. If successful, clinical trials will follow in patients with a single disease or with diabetes and complications. The aggregate effects on multimorbidity will likely only be explored after interventions are brought to market and are used by patients suffering from multimorbidity. It will then be important to monitor the effects across diseases. This will no doubt be complicated by various other medications already being taken.

## 9. Conclusions

This review offers a unique perspective on multimorbidity in COPD, a perspective in which AGEs and RAGE signaling constitute important common drivers and therapeutic targets.

It is demonstrated that the AGE-RAGE axis is disturbed in COPD and various comorbidities that are also non-communicable chronic inflammatory diseases. As reviewed, most of these disturbances could derive from common exposures and causes, including complex genetic influences and the metabolic context throughout life. Studies performed in subjects suffering from individual diseases and studies that use animal models of these diseases demonstrate that AGEs and RAGE play an important role in their pathogenesis. Across COPD and comorbidities, common pathogenic processes are induced by RAGE and AGEs, and this axis can execute important crosstalk between organ systems. Circulating levels of AGEs and sRAGE are further suggested as biomarkers for these individual diseases.

Yet, the biology of RAGE and AGEs is far from understood, and important questions remain, as highlighted in this review. However, because the clinical picture often includes multimorbidity, it is important that future research consider the contribution of AGEs and RAGE to the occurrence of multimorbidity in COPD, since it could be tackled by single therapeutics that target RAGE or AGEs.

## Figures and Tables

**Figure 1 jcm-12-03366-f001:**
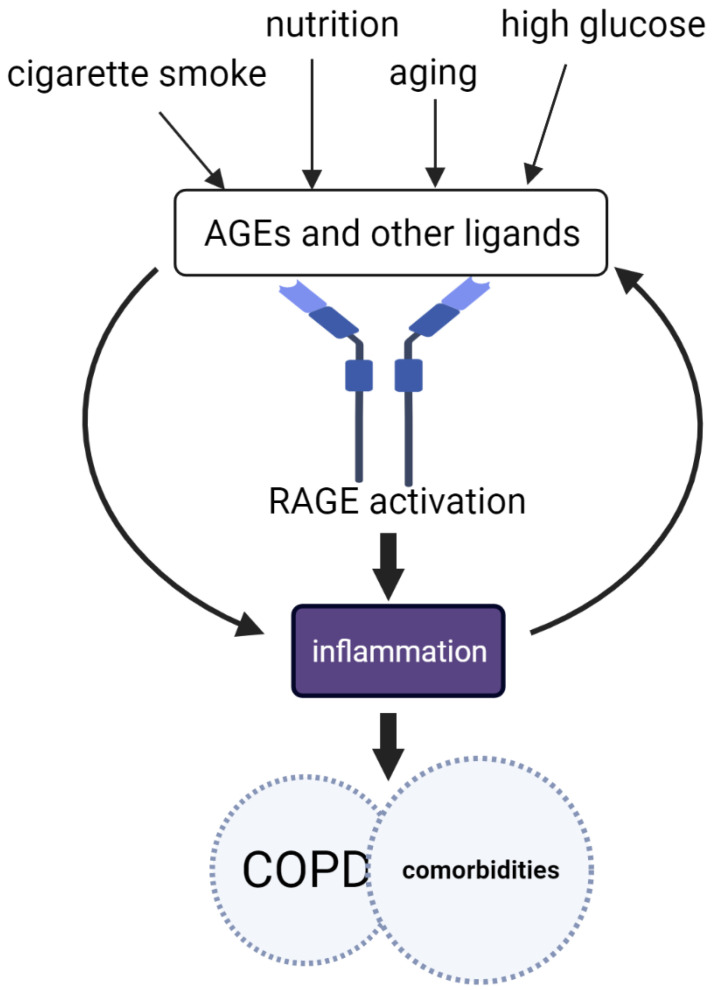
Simplified overview of sources of AGEs and other ligands that trigger RAGE signaling, leading to inflammation. Inflammation is sustained through the production of more ligands and RAGE and contributes to COPD and comorbidities. Figure created with Biorender.com (accessed on 24 April 2023).

**Figure 2 jcm-12-03366-f002:**
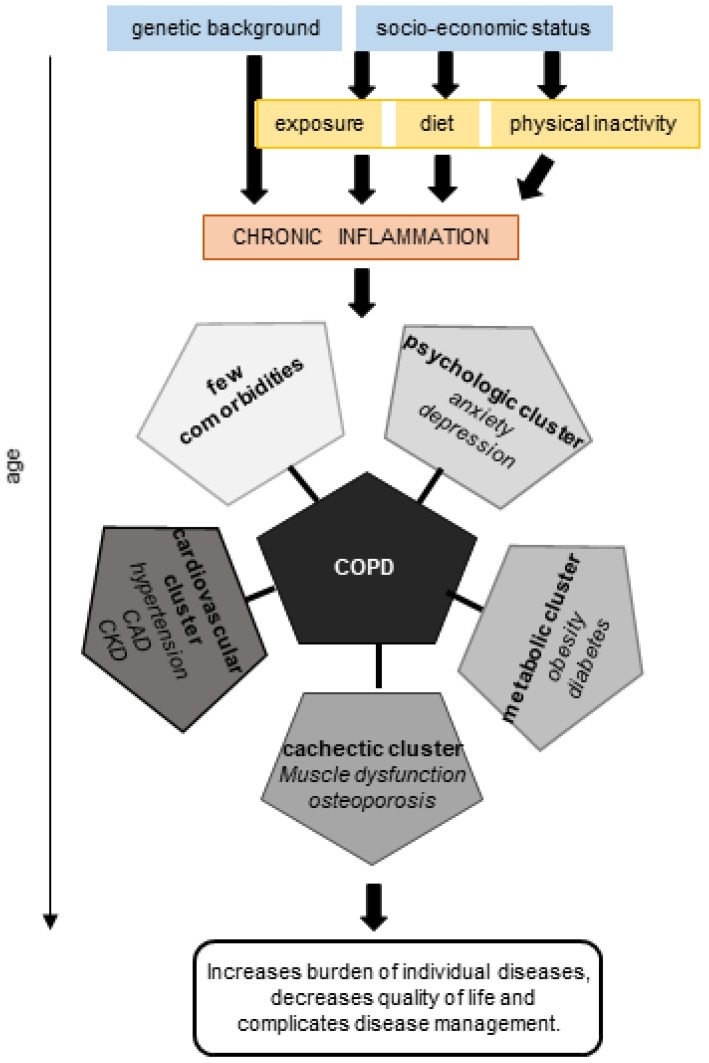
Multimorbidity in COPD is associated with common causes. Comorbidities in COPD occur in clusters, suggesting the involvement of common pathways in which chronic inflammation plays an important role.

**Figure 3 jcm-12-03366-f003:**
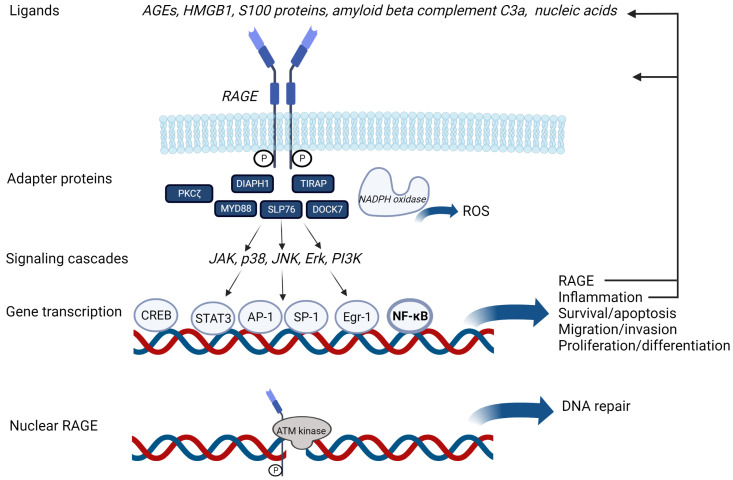
Complexity of RAGE signaling. A multitude of RAGE ligands exist that activate numerous intracellular signaling pathways through various adapter proteins recruited to the intracellular domain. Through increased gene transcription, RAGE activation leads to inflammation, survival or apoptosis, migration and/or invasion, and proliferation and/or differentiation. Signal amplification is achieved through enhanced expression of RAGE itself as well as of various RAGE ligands, in which activation of NAPDH oxidase and NF-κB play a role. RAGE can also be present in the nucleus, where it has been shown to contribute to the repair of DNA double-strand breaks. Figure created with Biorender.com (accessed on 24 April 2023).

**Figure 4 jcm-12-03366-f004:**
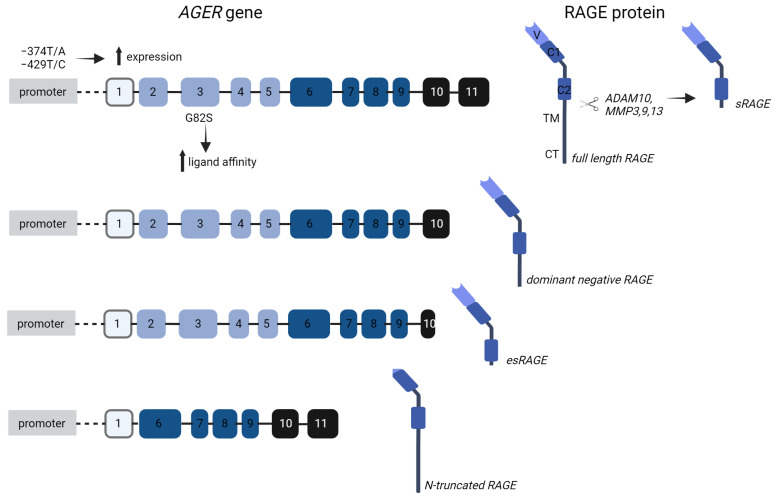
RAGE isoforms and genetic variants. The *AGER* gene contains 11 exons. The full-length RAGE protein consists of two constant domains (C1, C2) and a variable (V) immunoglobulin domain, which together are involved in ligand binding. A transmembrane domain (TM) elicits ligand-induced oligomerization, and a cytoplasmic domain (CT) interacts with the various downstream effectors. Full-length RAGE can be cleaved into soluble RAGE by ADAM10 and MMP3, 9, and 13. Alternative splicing gives rise to endogenous soluble RAGE (esRAGE), which lacks the TM and CT, a dominant negative (DN) variant that lacks the CT, and an N-truncated variant that lacks the V-domain. Polymorphisms in the *AGER* promotor increase *AGER* expression. A SNP leading to an amino acid substitution in the V domain (G82S) increases ligand affinity and downstream signaling. Figure created with Biorender.com (accessed on 8 May 2023).

**Figure 5 jcm-12-03366-f005:**
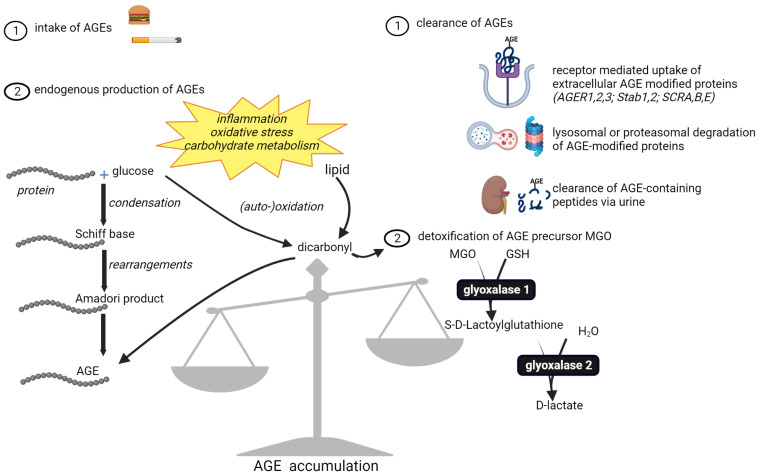
Causes of AGE accumulation. Dietary intake of AGEs and exposure to glycotoxins through cigarette smoke lead to AGE accumulation in tissues. Endogenous AGE formation starts with the condensation of a reducing sugar with a free amine group in a protein and proceeds through various chemical reactions and molecular rearrangements. It is accelerated under conditions of inflammation, oxidative stress, and hyperglycemia, which stimulate the formation of dicarbonyls. AGE-modified proteins are cleared by receptor mediated uptake and intracellular degradation through the ubiquitin-proteasome system, or autophagy. The resulting AGE-modified peptides are cleared from the body by the kidneys. The glyoxalase system detoxifies MGO, which is a dicarbonyl and an important AGE precursor. These clearance and detoxification systems are negatively impacted by aging and inflammation/oxidative stress. Figure created with Biorender.com (accessed on 24 April 2023).

**Figure 6 jcm-12-03366-f006:**
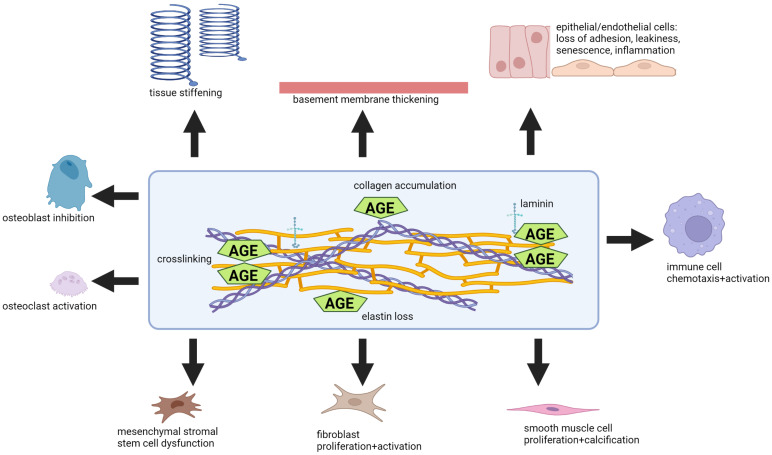
Overview of the pathophysiological effects of AGEs through modification of the extracellular matrix. AGE modification of ECM proteins leads to alterations in ECM turnover and tissue stiffness and influences various cells in contact with the ECM. Figure created with Biorender.com (accessed on 24 April 2023).

**Figure 7 jcm-12-03366-f007:**
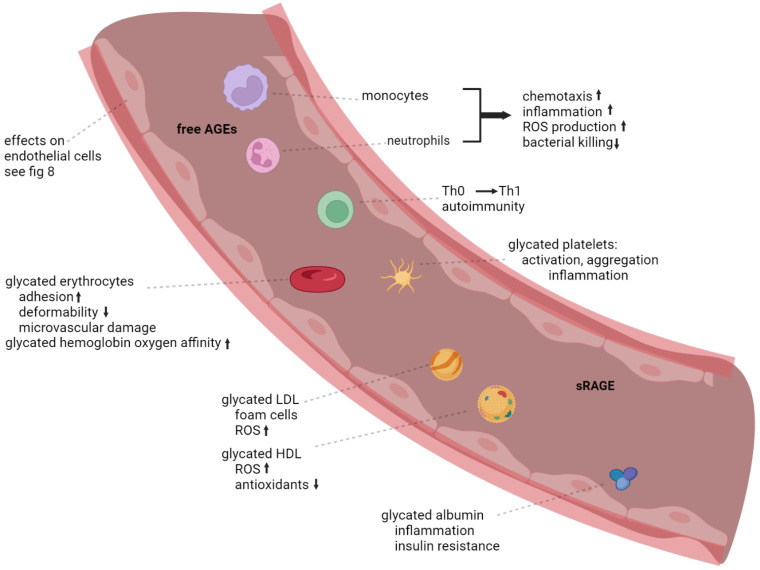
Overview of interorgan crosstalk mediated by circulating AGEs. Circulating AGEs affect blood and endothelial cells through the induction of RAGE signaling, whereas AGE modification also directly influences platelets and red blood cells, as well as the function of plasma proteins including HDL, LDL, and albumin. Figure created with Biorender.com (accessed on 24 April 2023).

**Figure 8 jcm-12-03366-f008:**
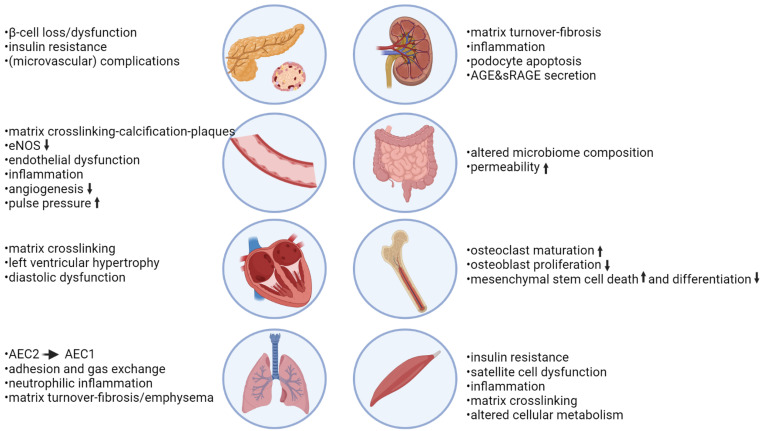
Overview of the effects of AGEs and RAGE on various organs and their potential contribution to disease development and progression. Figure created with Biorender.com (accessed on 24 April 2023).

**Figure 9 jcm-12-03366-f009:**
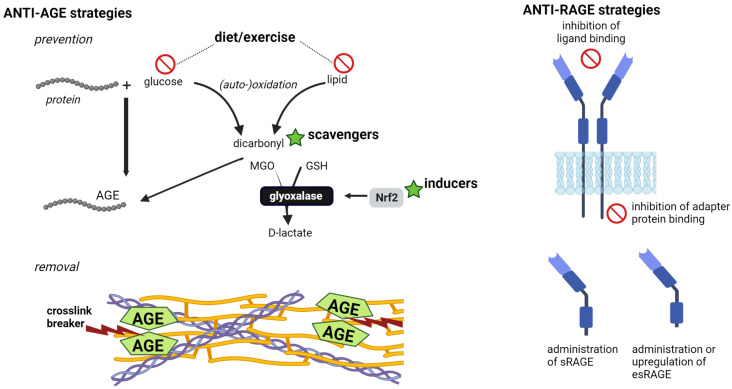
Overview of interventions aimed at AGE-RAGE. The left panel provides an overview of strategies that prevent the intake or formation of AGEs or remove AGE-induced crosslinks. The right panel provides an overview of various strategies that target RAGE. Figure created with Biorender.com (accessed on 8 May 2023).

**Table 1 jcm-12-03366-t001:** Overview of strategies and specific compounds that counteract or prevent AGE-RAGE.

Inhibition of AGEs	Cross-Link Breakers	Anti-RAGE
Limiting intake	*Alagebrium (ALT-711)*	Ligand binding inhibitors
Exercise		*FPS-ZM1*
Precursor scavengers		*Azeliragon (TTP488)*
*Aminoguanidine*		Interaction with adapter
*Pyridoxamine*		protein inhibitors
*Quercetin*		*DIAPH: RAGE299*
*Metformin*		*SLP76: TAT-SAM*
*Rosiglitazone*		sRAGE administration
Inducers of glyoxalase		esRAGE upregulation
*Curcumin*		
*Resveratrol*		
*Sulforaphane*		
*Hesperetin*		

## Data Availability

Not applicable.

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
