# Peer review of "The AGE-RAGE Axis and the Pathophysiology of Multimorbidity in COPD"

_jcm, 2023, doi:10.3390/jcm12103366_

Round 1
Reviewer 1 Report
The review is well-structured and relevant to the field. The authors referred that cigarette is the leading cause of COPD; wood smoke is another highly prevalent cause of this disease. There is a different pathophysiological pattern distinguishing COPD derived from cigarette smoke and that derived from wood smoke, what the authors have to say in this respect. Is there a different expression concerning AGER polymorphism or RAGE expression and concentration?. I enjoyed reading it.
In line 845, the word "to" is missing in the phrase "is thought be involved."
In line 880, add the letter "e" to the word, therefore
I recommend to check the typo
Reviewer 2 Report
The review of the role of RAGE in COPD and the multimorbidities associated with COPD is comprehensive and well written. The authors make a strong case for the association of AGERs and RAGE in inflammatory related diseases. I appreciate the inclusion of in vitro effects and animal models highlighting the effects of external environmental factors (e.g. cigarette smoke) on RAGE levels and expression. The figures are clear and helpful, size may have to be increased. I especially appreciate the inclusion of epigenetics. I wonder if the effect of human RAGE polymorphisms on disease could be expanded slightly. With regards to COPD (my research focuses on Pulmonary diseases) is the lack of discussion on COPD phenotypes (chronic bronchitis and emphysema) which may have distinct/overlapping mechanisms. Much research has now focused on "muco-obstructive respiratory disease" and a discussion on the effects of RAGE on mucins would be a positive inclusion.
Reviewer 3 Report
The authors present a comprehensive review of AGE-RAGE pathways in COPD and common comorbid conditions. While the topic is of significant interest, the readability and organization of the manuscript detract from its potential impact. This must be addressed prior to consideration for publication, along with specific comments below.
Major comments:
- Recommending revising the manuscript in general for clarity-lots of compound/complex sentences (seen throughout, however particular examples in the abstract/introduction-lines 34-44, 79-83, etc). Divide long paragraphs into more manageable chunks.
- The manuscript is very long with an extremely long list of references-Overall it would benefit from trimming and prioritization of key concepts and points. It often gets bogged down in detail/pathways/complex terminology (examples of this are seen throughout, e.g. lines 912-916; section 5.4).
- Rather than a comprehensive and complete listing of all evidence concerning AGE-RAGE across numerous diseases and organ systems, it would be more beneficial to provide a more streamlined and thematically-focused summary centered on the pathway’s importance in COPD. While multimorbidity/other diseases should be considered, the length and complexity of these sections distract from this core premise. In general, would benefit from tying various organ and pathway-specific sections together under this theme. Certain sections not directly related to COPD (e.g. 5.1, 5.2) could be reduced in length with summaries of evidence and key points- Beginning organ/comorbidity-specific sections with summary sentences that link biological pathways/extrapulmonary AGE-RAGE to COPD (e.g. lines 554-549) are useful and would serve to emphasize the central theme.
- Consider reorganizing the latter portions of the manuscript. After introduction and lung-specific sections, the review goes through comorbidities/extrapulmonary manifestations, then back into dense biological pathways (then back to systemic effects in section 7.3). Should these pathways be summarized/reviewed earlier (before lung/organ-specific effects of AGE-RAGE)?
Minor comments:
- Introduction (may integrate with a simplified version of figure 2/section 4.1): Simple figure/schematic for RAGE (sources of AGE, organ-specific effects, etc) could be helpful. Figure 2-emphasize positive feedback loop via NFkB
- Line 73-smoking cessation can slow, but may not prevent disease progression
- Lines 92-94: It would also be appropriate to mention cardiovascular disease as a key cause of mortality that is related to chronic inflammation (this is discussed later in the introduction but not included here)
- Figure 3: correct spelling of promoter. It would be helpful to link RAGE isoforms to site(s) of tissue expression in the figure
- Appreciate emphasis of early life events/lung development in COPD
- Figure 4 legend-anime groupàamine group
- Section 6.4-Move summary sentences in line 655; 664 earlier to point out opposing associations for sRAGE/esRAGE in DM/inflammation with a sentence introducing this idea-it is easy to overlook the subtle difference in these two acronyms and is confusing to read at first glance
- Section 6.4-Is there potential interaction of RAGE with corticosteroids in COPD? What is the effect on bone turnover/osteoporosis?
- Spelling-therefore, line 880
- Lines 1072-1075-may want to mention recent work on Nrf2/sulforaphane and macrophage phagocytic function in COPD
Round 2
Reviewer 3 Report
The authors have revised the manuscript and have addressed many of my comments on the original version. It is much more readable and while it remains detailed, is much easier to follow.